# High Bio-Content Thermoplastic Polyurethanes from Azelaic Acid

**DOI:** 10.3390/molecules27154885

**Published:** 2022-07-30

**Authors:** Bhausaheb S. Rajput, Thien An Phung Hai, Michael D. Burkart

**Affiliations:** Department of Chemistry and Biochemistry, University of California, San Diego, 9500 Gilman Dr., La Jolla, CA 92093-0358, USA; bsrajput@ucsd.edu (B.S.R.); h1phung@ucsd.edu (T.A.P.H.)

**Keywords:** bio-carbon content, polyester polyols, thermoplastic polyurethanes, prototyping

## Abstract

To realize the commercialization of sustainable materials, new polymers must be generated and systematically evaluated for material characteristics and end-of-life treatment. Polyester polyols made from renewable monomers have found limited adoption in thermoplastic polyurethane (TPU) applications, and their broad adoption in manufacturing may be possible with a more detailed understanding of their structure and properties. To this end, we prepared a series of bio-based crystalline and amorphous polyester polyols utilizing azelaic acid and varying branched or non-branched diols. The prepared polyols showed viscosities in the range of 504–781 cP at 70 °C, with resulting TPUs that displayed excellent thermal and mechanical properties. TPUs prepared from crystalline azelate polyester polyol exhibited excellent mechanical properties compared to TPUs prepared from amorphous polyols. These were used to demonstrate prototype products, such as watch bands and cup-shaped forms. Importantly, the prepared TPUs had up to 85% bio-carbon content. Studies such as these will be important for the development of renewable materials that display mechanical properties suitable for commercially viable, sustainable products.

## 1. Introduction

The field of renewable and biodegradable polymers is rapidly growing and attracting significant attention due to pressing environmental concerns [1,2,3,4,5,6,7,8]. Renewable feedstocks will play an important role in reducing greenhouse gas emissions and ensuring energy security in the 21st century [7,9,10]. Progressively increasing the renewable feedstock content of chemicals and materials will help create the mass markets for renewable feedstocks required to bring down their costs to make them competitive with petroleum sources [11]. The utilization of bio-based materials for product prototypes is a precondition to competing in the marketplace and represents a first step that can be met by material scientists. Thermoplastic polyurethanes (TPUs) are among the most versatile and broadly adopted materials used in applications such as shoes, industrial belts, sporting goods, medical devices, cabling, golf balls, cellular phone covers, watch bands, and automobile interiors. TPUs are segmented linear block copolymers formed by reaction of alternating soft segments (polyols) and hard segments (made from reaction of chain extenders and isocyanates) [12]. TPUs attract substantial interest because they possess many useful properties, including mechanical strength, modulable flexibility, good abrasion resistance, elasticity, and transparency [12]. Depending on the structure of the polyol and isocyanate during synthesis, this material can present properties that range from soft elastomers to hard plastics [13]. Most TPU products are made from petroleum-based chemicals, and utilizing bio-based ingredients for the development of high-performance TPUs is a major focus of current polyurethane research.

Dicarboxylic acids obtained from biological sources are important biomass-derived chemicals that are used for many monomers and industrial chemical intermediates [14]. As a result, it is projected that bio-derived carboxylic acids can be made economically viable, and may result in the development of disruptive bio-based aliphatic polyesters. Dicarboxylic acids are one of the main ingredients for making polyester polyols for polyurethane applications and are key building blocks for TPUs. Recently, aliphatic polyesters from bio-derived carboxylic acids have been explored for use in renewable TPUs [15,16,17]. Given that we can source azelaic acid from microalgae, the most sustainable photosynthetic biomass, we have focused our attention on developing polyurethanes from this monomer [18].

In a recent study of TPUs made from azelate polyols, it was shown that even-numbered methylene repeat units from n-alkanediols can result in desirable physical properties [19]. The combination of bio-based azelaic acid with succinic and adipic acids for the synthesis of co-monomeric polyester polyol soft segments has provided significant improvements in dynamic properties of TPUs [20]. A comparison between petroleum-based adipic versus renewable azelaic acid polyester polyols as building blocks for soft TPUs indicated a slightly higher degree of phase separation for azelate compared to analogous adipate polyols [16]. A study on structure–property relationships for crystalline and amorphous azelate polyols, and their effect on TPU properties, found that TPUs based on crystalline azelate polyols have higher material strength compared to amorphous azelate polyol-based TPUs [17]. In another report, the authors prepared segmented thermoplastic polyurethanes from various molecular weight polyester polyols with 1,6-hexamethylene diisocyanate (6HDI) and 1,3-propanediol (1,3-PDO) and investigated the degree of microphase separation [21]. To date, most of these TPUs have been prepared from aromatic diisocyanate (4,4′-MDI), and there is no report on the utilization of aliphatic diisocyanates, such as 1, 6-hexamethylene diisocyanate (6HDI) with bio-based azelate polyester polyols, for the preparation of TPUs. Similarly, no studies have reported the effect of amorphous azelate polyols incorporated with 2-methyl 1,3-propanediol (2MPDO), and 3-methyl 1,5-pentanediol (3MPDO) branched-chain diols on the structural and functional property relationship of TPUs.

Here, we explore the effect of two diols, 2MPDO, and bio-based 3MPDO, incorporated into amorphous azelate polyols and compare these with 1,3-propanediol-based crystalline azelate polyol. We demonstrate the usefulness of amorphous TPUs for prototype applications.

## 2. Results and Discussions

### 2.1. Polyester Polyol Synthesis

In the present study, bio-based azelate polyester polyols were synthesized from commercial azelaic acid and various branched and non-branched diols, including 1,3-propanediol, 2-methyl 1, 3-propanediol (2MPDO), and 3-methyl 1, 5-pentanediol (3MPDO), and using a dibutyltin dilaurate (DBTDL) catalyst (see Table 1 for details of the formulation). Although the combination of azelaic acid with 1, 3-propanediol-based polyester polyol is well explored in the literature [17], polyester polyols based on azelaic acid in combination with 2MPDO and 3MPDO have been not reported. Here, the synthesis of polyester polyols was carried out under a nitrogen atmosphere at 160 °C, where the rapid release of water condensate was observed. The temperature was raised to 180 °C to complete the conversion of diacid and diol into polyester polyol (see Section 3 for detailed polyol synthesis and Table 1 for detailed polyol formulations). The individual polyol reactions were continued until anticipated hydroxyl and acid numbers were achieved (for instance, below 2 mg KOH/g). The formation of the resulting polyester polyols was investigated using NMR spectroscopy (Figure 1). ^1^H NMR of azelate polyols such as AzAPDO, AzA2MPDO, and AzA3MPDO displays a chemical shift at around δ 2.08–2.24 ppm, which matches the ester group attached methylene protons (originated from the azelaic acid backbone, respectively), while the peak at δ 3.95–4.17 ppm corresponds to the ester-attached methylene group (originated from the 1,3-PDO, 2MPDO, and 3MPDO diols backbone) (Table 2, Run 1–3; ESI Appendix A). The structural identity of AzAPDO, AzA2MPDO, and AzA3MPDO polyols was further validated by ^13^C NMR. Resonance at δ 173.7–174.3 ppm is the distinguishing peak of an ester carbonyl carbon in polyester polyols (Table 2, Run 1–3; ESI Appendix A). The prepared polyester polyols were subjected to viscosity analysis, and all viscosity measurements were carried out at 70 °C. Due to the structural diversity in the obtained polyester polyols, all showed variable viscosity values. For example, crystalline AzAPDO had viscosity up to 683 cP, which closely matches the reported value [17], whereas the other two amorphous polyester polyols, AzA2MPDO and AzA3MPDO, displayed viscosity values up to 781 cP and 504 cP, respectively. The increased viscosity in the case of branched AzA2MPDO polyester polyol is explained by the presence of the branched methyl group, which restricts molecular motion due to increased polymer chain slip-resistance within similar molecules. These polyol viscosity observations were supported by a study conducted by TNM Tuan Ismail and co-workers, whereby the viscosity of a polyester polyol was found to be increased with an increase in the volume of branching in the molecule [17]. The lower viscosity observed for AzA3MPDO polyester polyol is most likely due to the increased chain length of diol (3MPDO), which ultimately increased the chain length of the resultant polyester polyol and hence decreased the physical interactions, such as polar interaction between the alcohol and ester functional groups. The crystalline polyester polyol (AzAPDO) was found to be solid at room temperature (25 °C), and amorphous polyols showed a liquid form (Table 2).

### 2.2. Synthesis of Thermoplastic Polyurethanes

Azelate polyester polyols were subjected to TPU synthesis (Figure 2). Prior to synthesis, polyols were dried in a vacuum oven at 70 °C for 24 h. TPU synthesis was performed at the low temperature of 75 °C in plastic cups using a speed mixer with an appropriate monomer ratio. Preheated polyol, chain extender (such as 1,3-PDO), and catalyst were premixed and added the preheated (at 75 °C) hexamethylene diisocyanate (6HDI), where the ratio between polyol mixture/NCO was 1/1.1 (see Table 3 for detailed formulation). The resulting mixture was poured into a Teflon dish to generate an average sheet thickness of 3–4 mm for further analysis. For real-world applications, the TPUs were also demonstrated for flexibility and toughness via a watch and plastic cup mold, as prototypes for high bio-content TPUs to replace petroleum versions (Figure 1).

The synthesized TPUs contained up to 85% bio-carbon content. The calculation of bio-carbon content is based on the following formula.
$$\mathrm{Bio}-\mathrm{carbon}\;\mathrm{content}\;\left(\%\right)=\frac{\mathrm{Weight}\;\mathrm{of}\;\mathrm{bio}-\mathrm{based}\;\mathrm{carbon}\;\mathrm{in}\;\mathrm{grams}}{\mathrm{Weight}\;\mathrm{of}\;\mathrm{total}\;\mathrm{carbon}\;\left(\mathrm{bio}-\mathrm{based}\;\mathrm{and}\;\mathrm{non}\;\mathrm{bio}-\mathrm{based}\right)\;\mathrm{in}\;\mathrm{grams}}\times 100$$

The formation of TPU was confirmed by FTIR spectroscopy. The complete disappearance of the free isocyanate peak at around 2250 cm^−1^ indicated the formation of TPUs (Figure 2). The peak at 3319–3324 cm^−1^ is assigned to the stretching vibration of hydrogen-bonded -NH moieties in the urethane groups of hard segments. In TPUs, C-H stretching vibrations in -CH_2_ groups as bimodal bands with maxima were observed in the range of 2926–2934 cm^−1^, respectively. Furthermore, a characteristic band between 1722–1730 cm^−1^ and 1682 cm^−1^ is associated with the stretching vibration of C=O in TPUs; such an observation was reported in the literature for thermoplastic polyurethanes [22]. The bands at 1535–1539 cm^−1^ were assigned to stretching vibration of -CN. The peak at 1722–1730 cm^−1^ corresponds to the free carbonyl group, whereas the peak at 1682 cm^−1^ is attributed to the hydrogen-bonded carbonyl group. In TPUs, the bending vibrations of -CH groups were registered between 1461 and 1465 cm^−1^. TPUs showed multiple IR bands in the range 1100–1200 cm^−1^, related to the C(O)-O-C stretching vibration from the ester groups of polyester polyols.

Furthermore, the structural identity of TPUs was ascertained from ^1^H and ^13^C NMR. In ^1^H NMR, the chemical shift of δ 6.99–7.07 ppm corresponds to urethane protons (-NHCO-), indicating the formation of TPUs (Figure 3, Figure 4 and Figure 5). All other backbone protons match with their respective chemical shifts. For example, the signal at around δ 0.87–0.88 ppm originates from branched methyl protons in TPU2, whereas in the case of TPU3, the peak at δ 0.83 ppm corresponds to branched methyl protons due to the 3MPDO diol. A clear observation was made from ^13^C NMR; the chemical shift at around δ 157 ppm further suggests the formation of urethane linkage (-NHCO-) due to the reaction of polyol mixture and isocyanate (Appendix A for TPU1 and TP2, respectively). The formation of high-molecular-weight TPUs was also confirmed with the help of gel permeation chromatography (GPC). The TPUs displayed a weight average molecular weight (M_w_) of between 138 × 10^3^ and 179 × 10^3^ g/mol, and a polydispersity index (PDI) in the range of 2.82–3.36 confirmed the broad distribution of polymeric chains.

### 2.3. Properties of Thermoplastic Polyurethanes

Thermal properties of TPUs were evaluated using differential scanning calorimetry (DSC) and thermal gravimetric analysis (TGA). DSC analysis of TPUs was carried out between −120 °C and 220 °C under a nitrogen atmosphere. All TPUs displayed clear glass transition temperature values (T_g_s) in low-temperature regions (−65 °C to −53 °C), which are related to the azelate polyester polyols soft domains (Figure 6). The T_g_ values were observed in the order of TPU1 > TPU2 > TPU3, with the lower T_g_ value associated with a chain length of polyols in TPUs. For example, longer chain length AzA3MPDO polyol-based TPU3 showed a lower T_g_ of −65 °C than shorter chain length TPU1 (T_g_ = −53 °C; based on AzAPDO polyol). Comparatively, the higher T_g_ in the shorter chain length polyol is presumably because of increased hydrogen bonding between azelate polyol soft segments and hard segments (1,3-PDO- and 6HDI-based urethane linkage), which basically decreased the mobility of the azelate polyol chains in TPU1 and TPU2 compared to TPU3. Interestingly, TPU1 made from AzAPDO polyol displayed two melting temperatures, T_m1_ and T_m2_, at 45 °C and 105 °C, respectively. However, a sharp T_m_ was observed at 45 °C, which stems from the AzAPDO crystalline soft segments, and T_m_ at 103 °C is because of the hard segment, whereas the amorphous TPUs, TPU1, and TPU2, displayed T_m_ of 105 °C and 104 °C, respectively. The crystallization temperature (T_c_) was also observed for TPUs. TPU1 based on AzAPDO polyols showed two distinct crystallization temperatures, T_c1_ and T_c2_, at 19.1 °C and 70.7 °C, respectively, whereas TPU2 displayed T_c_ at 38.2 °C and TPU3 displayed T_c_ at 54.8 °C. The crystalline TPU1 displayed a higher shore A hardness compared to amorphous TPUs due to the branching structure of TPU2 and TPU3 (Table 4). The hardness of our TPUs entirely depends on the microphase separation of TPUs due to diverse polyols structures; in the case of TPU1, the microphase separation results in higher hardness compared to TPU2 and TPU3, which have lower microphase separation. A similar observation was made in other studies, whereby crystalline azelate polyols were found to be harder than amorphous ones [17].

In TGA analysis, all prepared TPUs showed thermal stability up to 280 °C. Thermal decomposition was observed above this (Figure 7A). The TPU1 and TPU2 showed 29% weight loss at the temperature of 376 °C and 386 °C, respectively, whereas maximum weight loss of around 38% was observed for TPU3 at just 378 °C, confirming the weaker physical interactions of polymer chains in TPU3. DTG curves showed a thermal decomposition at 316 °C, 330 °C, and 336 °C corresponding to the urethane group, and thermal decomposition at 410 °C, 412 °C, and 422 °C originating from ester linkage breakdown at elevated temperatures. The shoulder peak was also observed in the range of 460–465 °C associated with hydrocarbon chain decompositions (Figure 7B). Such thermal decomposition for functional polyurethanes was reported in the literature [23]. Mechanical properties were examined using a dynamic mechanical analyzer (DMA) and universal testing machine (UTM). DMA analysis of TPU1 showed a gradual decrease in storage modulus because of the crystalline nature of the resultant TPU1, which provides more strength in TPU1 compared to TPU2 and TPU3 (Figure 8A). On the other hand, a sharp decrease in storage modulus was noted for amorphous TPU2 and TPU3. Figure 8B represents the DMA data obtained from tan delta versus temperature curves. The T_g_ obtained from the DMA curves for TPUs showed a value in the range of −33.2 to −43.8 °C, which correlated with T_g_ values obtained from DSC measurement. The lower T_g_ values by DMA analysis further suggest that the TPU possesses soft block structures due to polyols. The TPU1 was found to dissipate less energy due to its crystalline nature. On the other hand, the increasing tan delta in TPU2 and TPU3 (due to their amorphous and more elastic nature) indicates that the material has more energy dissipation potential, so the greater the tan delta, the more dissipative the material at a given applied oscillatory force. The DMA analysis observation was supported by tensile strength measurements. The crystalline TPU1 exhibited the highest tensile strength (48 MPa), which correlated to strong hydrogen bonding or physical interactions between the polymeric chains. In contrast, the amorphous TPUs have a disrupted packing between the polymeric chains due to the presence of a side methyl group (Table 3). Moreover, lower tensile properties for amorphous/branched TPUs are related to the decrease in the degree of microphase separation [18]. After TPU1 (48 MPa), TPU2 was found to have a the next-highest tensile strength (31 MPa), and TPU3 was the lowest (18 MPa) (see the stress–strain curve for more details, Figure 9). This is most likely due to the longer chain length structure of TPU3 exhibiting weaker hydrogen bonding between the polymeric chains, resulting in the lower tensile strength value. The crystalline TPU1 had a lower elongation at the break in contrast to the amorphous TPUs. This is due to the presence of a side methyl group in amorphous azelate polyols, resulting in the disruption of crystallization of the soft segment chains [24].

## 3. Materials and Methods

### 3.1. Materials

Azelaic acid (Crodacid DC1195, 100% bio-based) with 95% purity was supplied by Croda Inc. The 1,3-propanediol (bio-based) was obtained from Susterra and used as received for polyol and TPU preparations. The 2-methyl 1, 3-propanediol was received from Sigma Aldrich and used without further purifications. The bio-based 3-methyl 1, 5-pentanediol was obtained from Visolis as a gift sample and used as such without purifications. The dibutyltin dilaurate catalyst (DBTDL, 95%) was purchased from Sigma-Aldrich and used without purification. Hexamethylene diisocyanate (6HDI, 98%) was supplied by Alfa Aesar and used without purification for TPU synthesis. Sigma-Aldrich supplied 1.0 M tetrabutylammonium hydroxide in methanol and *p*-toluenesulfonyl isocyanate (96%) for hydroxyl number and acid number titration. HPLC grade toluene, 2-propanol acetonitrile, reagent grade 1-octanol, and potassium hydroxide were supplied by Fisher Chemical for hydroxyl and acid number analysis.

### 3.2. Measurements

FTIR analysis was carried out on a Perkin Elmer Spectrum X fitted with a ZnSe 1 mm ATR cell. Sixteen scans were taken at 1.0 cm^−1^ resolution. Proton NMR and carbon NMR spectra were recorded on a JOEL ECA 500. Dynamic mechanical analyzer (DMA) measurement for TPUs was performed out on a TA instrument with a DMA oscillatory temperature ramp using a 3-point bending clamp in the temperature range of −120 to 120 °C. The hydroxyl number and acid number titrations were carried out according to ASTM 1899 and D664, respectively. Viscosity measurement for polyols was carried out at 70 °C using a discovery hybrid rheometer 30 (HR 30) instrument. For tensile strength and elongation at break measurement, a universal testing machine (UTM) machine, AGS-X 20KN, was used at the rate of 100 mm/min. Differential scanning calorimetry (DSC) analysis was performed on a TA instrument from −120 °C to 220 °C at the rate of 10 °C/min under a nitrogen atmosphere. Thermal gravimetric analysis (TGA) was carried out on TA instrument from 50 °C to 900 °C using a temperature ramp of 10 °C/min in a nitrogen atmosphere. The gel permeation chromatography (GPC) technique is used to calculate the molecular weight and molecular weight distribution (polydispersity index) of the polymers. The molecular weight of TPUs was determined relative to a polystyrene standard and DMF served as the polymer solvent. The flow rate for GPC is 0.35 mL/min and the temperature is 60 °C.

### 3.3. Synthesis Method for Polyester Polyols

Polyester polyol synthesis was carried out in a three-necked round bottom glass reactor equipped with a Dean–Stark apparatus, reflux condenser, and an oil bath on a hot plate. Then, under a nitrogen atmosphere, calculated amounts of diacids and diols were added to the reactor (see Table 1 for detailed polyol formulation). The polyester polyol synthesis was started at 150–160 °C and the temperature increased up to 180 °C over a period of 1 h. However, a rapid release of water by-product was observed in the initial 4–5 h. Afterwards, the DBTDL catalyst was added when about 80% water was collected, and the reaction was run until the desired acid and OH number (reactions took approximately 2–3 days) were achieved. The progress of the polyester polyol reactions was monitored by analyzing the acid and hydroxyl numbers in a regular interval of time.

### 3.4. Synthesis Method for TPUs

Polyester polyols and 1,3-propanediol were dried in a vacuum oven for 24 h before polymerization. Then, polyester polyols, 1,3-propane diol as a chain-extender, and catalyst (DBTDL) were weighed into a plastic cup and heated to 75 °C, and the polyol mixture was mixed using the speed mixer (FlackTek, DAC 150.1 FVZ-K) at 2000 rpm. The preheated hexamethylene diisocyanate (6HDI) at 75 °C was then added into the polyol mixture and speed mixed at 2000 rpm for about 1 min. Afterward, the reaction mixture was poured into a Teflon petri dish before a gel point and subsequently cured at 75 °C for 2 days to obtained a desired TPU sheet. Thermal and mechanical properties were performed after one week of room temperature curing. For tensile testing, a dog bone-shaped cutting die was used following the ASTM D638 standard to make sample specimens.

## 4. Conclusions

Here, we report new high bio-carbon content TPUs from both crystalline and amorphous azelate polyester polyols. The synthesized polyester polyols were characterized by proton and carbon NMR spectroscopy. Prepared polyester polyols showed low viscosity behavior at 70 °C, which allowed us to process TPU synthesis at lower temperatures. The TPUs were molded into simple prototypes, i.e., cups and watch bands. The formation of TPUs was ascertained from FTIR analysis, which clearly showed the complete disappearance of the isocyanate peak around 2250 cm^−1^. Proton and carbon NMR analysis also revealed the formation of TPUs. All prepared TPUs displayed adequate thermal and mechanical properties. The TPU synthesized from crystalline azelate polyester polyol was found to be the mechanically strongest and toughest material compared to the TPUs derived from amorphous polyols, as confirmed by tensile strength and storage modulus analysis. These TPUs could be promising candidates for a real-world applications, where the properties and end-of-life scenarios are carefully tuned for use and disposal. These TPUs can be used in the application of making watch bands, mobile case covers, toys, industrial belts, etc. It is through the careful study of renewable materials such as these, and their demonstration as prototypes, that environmentally friendly plastics can find real-world applications.

## Data Availability

The data presented in this study are available on request from the corresponding author.

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
