# Peer review of "High Bio-Content Thermoplastic Polyurethanes from Azelaic Acid"

_molecules, 2022, doi:10.3390/molecules27154885_

Round 1

Reviewer 1 Report

Overall, this is a well-written manuscript, describing the synthesis and characterization of thermoplastic polyurethanes with high renewable carbon contents. However, before being published a few aspects need to be considered.

- Authors mentioned in the introduction part that "there is no report on utilization of aliphatic diisocyanates, such as 1, 6-hexamethylene diisocyanate (6HDI) with bio-based azelate polyester-polyols, for the preparation of TPUs." It is true, but it is worth mentioning that Saralegi et al. studied the effect of polyester polyol cristallinity on the properties of TPUs synthesized with HDI (https://doi.org/10.1002/pi.4330). The mentioned study has similarities with this study, so reviewer considers that authors should cite and mention this work in the introduction.

- The description of the FTIR spectra must include a deeper study of the carbonyl region, highlighting the free and hydrogen bonded urethane and carbonyl groups, in order to justify the phase separation and formation of segmented polyurethanes, as well as to see the differences in this behavior with the use of amorphous and crystalline polyols.

- Regarding thermal properties, authors should include DSC cooling and second heating scans, in order to study the thermal behavior after removing the thermal history of the samples.

- TGA derivative curves must be also presented in the manuscript, in order to understand better the degradation of the synthesized TPUs and the effect of soft segment crystallinity on the thermal stability,

- DMA curves must be presented with the y-axis in log scale, and tan delta curves must be also presented within the same graph. With these changes, authors should further describe the thermomechanical behavior.

- Finally, regarding tensile test, authors must include stress-strain graphs, since the shape of the curves can provide additional information about the mechanical behavior of the samples.

Reviewer 2 Report

The work “High Bio-Content Thermoplastic Polyurethanes from Azelaic Acid” deals with the synthesis and characterization of thermoplastic polyurethanes from polyols based on azelaic acid. The motivation for this work includes pro-ecological aspects related to the synthesis of new bio-based polyester polyols, which is essential these days. In my opinion, this work needs major revision. This paper seems to be incomplete.

1.       Lines 82-86 should be rewritten.

2.       Improve the quality of the Figures. One format style should be used.

3.       In Figure 2 please separate all curves, in the present version, this picture is illegible. Correct the axis y, the highest value should be 100%. Mark in the picture wavenumber of specific for TPU functional groups.

4.       It is better to preheat HDI at a lower temperature than 75 °C. This diisocyanate is a highly volatile substance and hazardous. This temperature can cause changes in the resulting molar ratio of NCO and OH groups.

5.       Please, explain the effect of the morphology of TPU on hardness, and add it to the text.

6.       Figure 6 should be improved. The scale on the X-axis should be limited to a range of temperature of measurement. Figure 8 should be improved. Please use the logarithmic scale. Some examples of how to do this can be found here: https://doi.org/10.3390/ijms22147438, https://doi.org/10.3390/polym14102033, https://doi.org/10.1016/j.eurpolymj.2019.109422, https://doi.org/10.1016/j.eurpolymj.2021.110673

7.       Extended analysis of TGA, DMA, and DSC results is needed.

8.       Add the Figure which presents the tanged delta vs temperature. Please compare the Tg of SS with the Tg determined based on DSC.

Reviewer 3 Report

The manuscript entitled “High Bio-Content Thermoplastic Polyurethanes from Azelaic Acid” aims to explore the effect of two diols incorporated into amorphous azelate polyols and compare these with 1,3-propanediol based crystalline azelate polyol. Also, the authors demonstrated the application of amorphous thermoplastic polyurethanes (TPUs) for prototype applications.

The manuscript is very well prepared. This topic is of high importance since the production of biodegradable polymers and plastic recycling are among the major concerns nowadays.

The introduction is good.

The materials and Methods are informative.

Results and Discussion are good but could be improved in my opinion. The authors could suggest more real-life applications of synthesized materials.

The conclusion is good but could also contain the idea of the precise suggestion for applying these materials.

The literature is up to date.

Round 2

Reviewer 1 Report

Authors should present only one stress-strain curve for each sample, and not all the curves performed.

Regarding DMA results, for TPU-1 sample, in tan delta curves, the damping factor due to the cristallinity of the sample is clearly observed, authors should comment that within the test.

Finally, reviewer suggests to present two graphs for TGA and DTG results, a and b, respectively. Authors should put together all the TGA curves in one graph and all the DTG curves in another graph, in order to see much clearer the differences between TPU samples.

Reviewer 2 Report

After the first revision, paper entitled “High Bio-Content Thermoplastic Polyurethanes from Azelaic Acid” reached a new quality. Still, some corrections are needed.

1.       Tensile test:  How many samples were used for the tensile test and calculation of average tensile strength and elongation at break?? This information should be given.

2.       Tensile test: Analyzing the tensile curves I suppose that samples TPU 1 and TPU were slipped out from the grips, isn't it?

3.       Something is strange with DMA curves. Storage modulus: I suppose that the unit on the OY axis should be MPa. Authors wrote: “DMA analysis of TPU1 showed a gradual decrease in storage modulus because of the crystalline nature of the resultant TPU1” – should be improved by e.g. XRD

4.       DSC curves: axis OX can be limited to the temperature range from -80 to + 160 deg. of C.

5.       Please add DSC curves of polyols  (for better justification of DSC results of TPU).

6.       TGA: On the DTG curves there is a third small peak (sample TPU 1 and TPU3).
